# A G-protein activation cascade from Arl13B to Arl3 and implications for ciliary targeting of lipidated proteins

**Katja Gotthardt[1†], Mandy Lokaj[1*†], Carolin Koerner[1], Nathalie Falk[2], Andreas Gießl[2], Alfred Wittinghofer[1*]**

[1]Structural Biology Group, Max Planck Institute of Molecular Physiology, Dortmund, Germany; [2]Department of Biology, Animal Physiology, University of Erlangen-Nuremberg, Erlangen, Germany

**Abstract** Small G-proteins of the ADP-ribosylation-factor-like (Arl) subfamily have been shown to be crucial to ciliogenesis and cilia maintenance. Active Arl3 is involved in targeting and releasing lipidated cargo proteins from their carriers PDE6$\delta$ and UNC119a/b to the cilium. However, the guanine nucleotide exchange factor (GEF) which activates Arl3 is unknown. Here we show that the ciliary G-protein Arl13B mutated in Joubert syndrome is the GEF for Arl3, and its function is conserved in evolution. The GEF activity of Arl13B is mediated by the G-domain plus an additional C-terminal helix. The switch regions of Arl13B are involved in the interaction with Arl3. Overexpression of Arl13B in mammalian cell lines leads to an increased Arl3·GTP level, whereas Arl13B Joubert-Syndrome patient mutations impair GEF activity and thus Arl3 activation. We anticipate that through Arl13B's exclusive ciliary localization, Arl3 activation is spatially restricted and thereby an Arl3·GTP compartment generated where ciliary cargo is specifically released.

**\*For correspondence:** mandy. miertzschke@mpi-dortmund.mpg. de (ML); alfred.wittinghofer@mpi-dortmund.mpg.de (AW)

[†]These authors contributed equally to this work

**Competing interests:** The authors declare that no competing interests exist.

## Introduction

Primary cilia are highly conserved organelles essential for developmental signalling pathways and cellular homeostasis. The small G-proteins of the Arl family Arl3, Arl6 and Arl13B have been shown to be important in the trafficking of ciliary proteins and structural integrity of the cilium (*Li et al., 2012*). Mutations in Arl proteins or their regulators can lead to cilia dysfunction causing ciliopathies such as Joubert syndrome (JS), Bardet–Biedl syndrome (BBS), or retinitis pigmentosa (RP) (*Cantagrel et al., 2008*; *Chiang et al., 2004*; *Schwahn et al., 1998*). Different ciliopathies are characterized by overlapping phenotypes such as renal cysts, polydactyly, brain malfunction, situs inversus, and vision impairment (*Waters and Beales, 2011*). Mutations in Arl6 –the first member of the Arl family found mutated in a human ciliopathy – cause BBS, whereas mutations in Arl13B lead to JS. JS in particular is characterized by a brain malformation with a characteristic molar tooth sign combined with polydactyly and kidney cysts. Although no mutations in Arl3 have been identified so far in ciliopathies Arl3$^{(-/-)}$ mice exhibit a ciliopathy related phenotype and die by 3 weeks of age (*Schrick et al., 2006*). One of the X-linked RP genes is RP2, which functions as a GTPase activating protein (GAP) specific for Arl3 (*Veltel et al., 2008*).

As most small G-proteins Arl3 cycles between inactive GDP-bound and active GTP-bound forms and in the latter it binds specifically to effectors (*Cherfils and Zeghouf, 2013*). Effectors of Arl3 are the carrier proteins PDE6$\delta$, which binds farnesylated and geranylgeranylated cargo, and Unc119a/b, which binds myristoylated cargo. Binding of activated Arl3 to the cargo-carrier complex induces conformational changes leading to the release of the cargo (*Ismail et al., 2012, 2011*; *Wright et al., 2011*). A close structural homologue of Arl3 is Arl2, which binds to the same set of effectors

**eLife digest** Most types of cells in humans and other animals have slender, hair-like structures known as cilia that project out of the cell surface. These structures sense and respond to signals from the external environment and are crucial for organisms to develop normally. Defects in cilia can lead to many serious conditions such as Joubert syndrome, which affects the development of the brain and other organs in humans.

The Arl family of "G-proteins" play important roles in the formation and operation of cilia. They contain a section called a G-protein domain whose activity can be switched on by interactions with other proteins called guanine nucleotide exchange factors (or GEFs for short). A member of the Arl family called Arl3 is found in higher amounts in cilia than in other parts of the cell. It is involved in the transport of proteins to the cilia from other parts of the cell, but it is not known which GEFs are able to activate it.

Here, Gotthardt, Lokaj et al. used several biochemical techniques to show that another member of the Arl family called Arl13B actually acts as a GEF to activate Arl3 in cilia. Arl13B is only found in cilia and the GEF activity relies on its G-protein domain and another element at one end called a C-terminal helix. Previous studies have shown that mutations in the gene that encodes Arl13B can cause Joubert syndrome in humans. Gotthardt, Lokaj et al. found that mutant forms of Arl13B had significantly lower GEF activity than normal Arl13B proteins.

Together, Gotthardt, Lokaj et al.'s findings provide an explanation for why Arl3 is only activated in cilia even though it is found throughout the cell. Further work is needed to understand how the activity of Arl13B is regulated.

(*Van Valkenburgh et al., 2001*). However, while Arl2 and Arl3 can release cargo such as Ras or RheB, only Arl3 is able to release ciliary cargo such as INPP5E, NPHP3, and GNAT-1 (*Ismail et al., 2012*; *Thomas et al., 2014*; *Wright et al., 2011*). The highly conserved Arl3 – only present in ciliated organisms – localizes throughout the cell and is enriched in the primary cilium (*Avidor-Reiss et al., 2004*; *Blacque et al., 2005*; *Zhou et al., 2006*). While RP2 functions as an Arl3 GAP and is thus important for the import of lipidated cargo by recycling Arl3 and its effectors (*Schwarz et al., 2012*; *Wright et al., 2011*; *Zhang et al., 2015*), the guanine nucleotide exchange factor (GEF) that activates Arl3 remains unknown. We had anticipated that in order for Arl3 to mediate cargo release inside cilia, an Arl3-specific GEF should be localized there as well.

## Results

To identify regulatory proteins of Arl3 we employed a yeast-2-hybrid (Y2H) screen using the fast cycling mutant Arl3$\triangle$N$^{D129N}$ as bait. The homologous mutation which in Ras was shown to decrease nucleotide and to increase GEF affinity while maintaining its ability to bind to effectors was used by us to identify the GEF for the plant specific ROP proteins (*Berken et al., 2005*; *Cool et al., 1999*). Screening a mouse retinal cDNA Y2H library identified several clones growing on selective media. Sequence analysis revealed Arl13B (residues 1–270) in addition to known Arl3 effectors such as PDE6$\delta$ and Unc119a. Intriguingly, in a parallel screen with mouse Arl13B$^{20-278}$ as bait Arl3 was found as rescuing clone. The interaction between Arl3 and Arl13B was further verified by directed 1:1 Y2H analysis (*Figure 1A*). Arl13B is an unusual Arl protein containing a C-terminal coiled-coil and proline rich region in addition to its G-domain (*Figure 1C*). The data show that the interaction is mediated by Arl13B's G-domain and part of the coiled-coil region. To investigate the specificity of the Arl13B-Arl3 interaction we tested the related constructs of Arl2 and Arl6 neither of which enabled growth on selective medium (*Figure 1B*).

To verify this interaction in vitro, we tested purified proteins in a glutathione-S-transferase (GST) pull-down assay. Due to better stability and purity of Arl13B from *Chlamydomonas reinhardtii* (*Cr*) the following experiments were performed with the homologous *Cr*-proteins purified from *Escherichia coli*. We thus prepared *Cr*Arl13B$^{18-278}$ (*Cr*Arl13B from now) analogous to mouse Arl13B$^{18-278}$ used in the Y2H screen and tested its interaction with GST-*Cr*Arl3 loaded with either

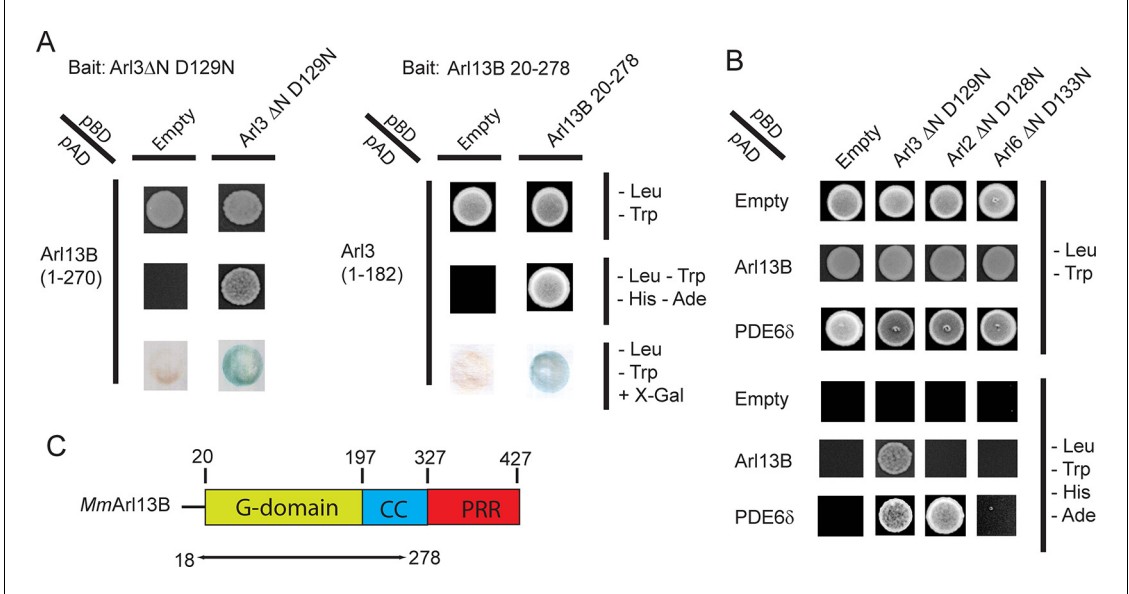

**Figure 1.** The interaction between ADP-ribosylation-factor-like (Arl) 13B (Arl13B) and Arl3 was identified in a yeast-2-hybrid (Y2H) screen. (**A**) Y2H interactions between Arl3ΔN D129N -pBD and Arl13B 1-–270-pAD and between Arl13B 20- – 278-pBD and Arl3-pAD. Transformed and mated cells were grown on –Leu –Trp medium. Interaction was verified on high stringency plates (-– Leu –Trp –His –Ade) and with a β-galactosidase filter assay. (**B**) Interaction of Arl13B 1-–270-pAD with Arl3ΔN D129N-pBD, Arl2ΔN D128N-pBD and Arl6ΔN D133N-pBD was analyzed on low and high stringency plates. PDE6δ-pAD was used as positive control for Arl3 and Arl2. (**C**) Domain architecture of Arl13B, numbering derived from murine Arl13B (*Mm: Mus musculus*).

GDP or GppNHp (a non-hydrolyzable GTP analogue) (*Figure 2A*). *Cr*Arl13B·GppNHp but not *Cr*Arl13B·GDP bound to GST-–*Cr*Arl3, and binding was slightly stronger to GST-*Cr*Arl3·GDP than to GST-*Cr*Arl3·GppNHp. These data show that the interaction between Arl3 and Arl13B is conserved between mouse and *Chlamydomonas*.

Since the rather weak interaction in the pull-down experiments and the nucleotide-independent binding suggested that Arl3 is not an effector for Arl13B, we turned our attention to a possible GEF function. As a ciliary protein Arl13B is a good, albeit, as a G-protein, a very unusual candidate GEF for Arl3. The dissociation of a fluorescent GDP-analogue (mantGDP) from *Cr*Arl3 in the presence of excess of unlabelled GTP was monitored after adding *Cr*Arl13B·GTP. The nucleotide dissociation was strikingly accelerated in the presence of *Cr*Arl13B·GTP and was dependent on the *Cr*Arl13B concentration (*Figure 2B* and *Table 1*). Consistent with features of a typical GEF (*Bos et al., 2007*) Arl13B did not discriminate whether mantGppNHp or mantGDP was bound to Arl3 and exchanged both nucleotides with the same velocity (*Table 2*). As a control *Cr*Arl6 did not stimulate the nucleotide release of *Cr*Arl3 nor did *Cr*Arl3 catalyze that of *Cr*Arl113B (*Figure 2B, C*).

We next asked whether the nucleotide-bound state of *Cr*Arl13B affects its GEF activity as suggested by the GST pull-down experiments. *Cr*Arl13B preloaded with GDP, GTP, or GppNHp was used to analyze the exchange activity. At 5 µM GEF, the exchange was about ninefold slower for GDP- than for GTP- and GppNHp-bound *Cr*Arl13B (*Figure 2E*). The observed rate constants of mantGDP-dissociation showed a hyperbolic dependence on *Cr*Arl13B concentration, with a maximum release rate of $0.86 \times 10^{-2}$ sec. The $K_M$ for the reaction is 1.1 µM for *Cr*Arl13B·GTP and 155 µM for *Cr*Arl13B·GDP (*Figure 2F*) showing that *Cr*Arl13B·GTP has a higher affinity than *Cr*Arl13B·GDP. Since the in vitro determined maximal nucleotide release stimulation of 70-fold appears relatively slow but not unusual, it is quite conceivable that additional factors such as the presence of membranes or lipids enhance the GEF activity as shown for the Ras-GEF SOS (*Gureasko et al., 2008*) and other GEFs (*Cabrera et al., 2014*; *Pasqualato et al., 2002*). Since Arl3 has a high affinity to membranes (*Kapoor et al., 2015*) and Arl13B is palmitoylated (*Cevik et al., 2010*) the reaction between them is thus most likely orchestrated on the ciliary membrane.

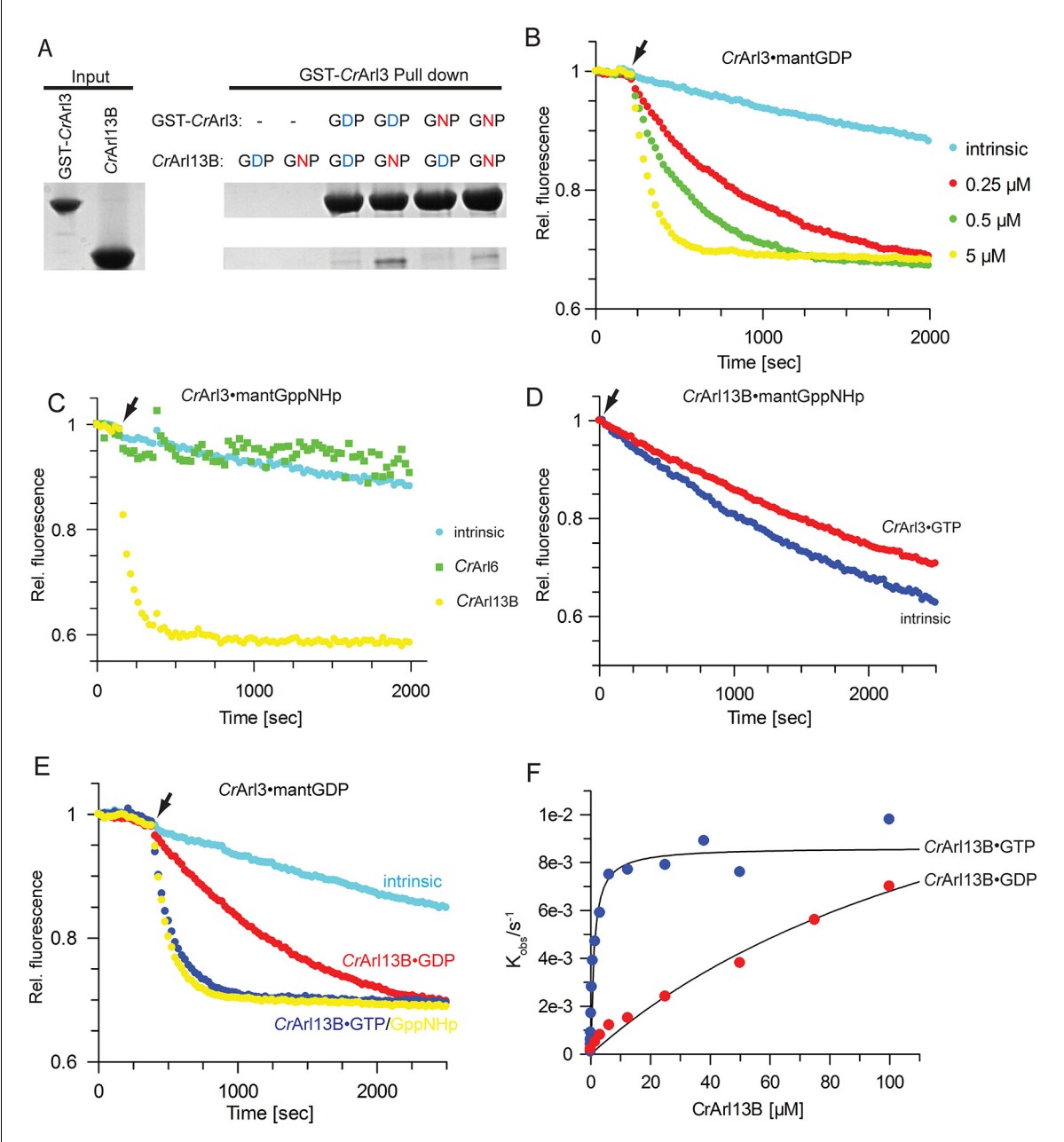

**Figure 2.** *Cr*Arl13B is the guanine nucleotide exchange factor for *Cr*Arl3. (**A**) Glutathione-S-transferase (GST) pull-down assay with purified *Chlamydomonas reinhardtii* Arl proteins as indicated and described in detail in Material and methods. (**B**) Guanine nucleotide exchange factor (GEF) activity of the indicated concentrations of *Cr*Arl13B[18–278] for 500 nM *Cr*Arl3·mantGDP. Arrow designates addition of *Cr*Arl13B and excess of unlabeled nucleotide. (**C**) *Cr*Arl13B·GppNHp but not *Cr*Arl6·GppNHp stimulates the nucleotide release of *Cr*Arl3·mantGppNHp. (**D**) *Cr*Arl3·GTP does not accelerate the nucleotide dissociation of *Cr*Arl13B·mantGppNHp. (**E**) GEF activity of 5 μM *Cr*Arl13B[18-278] loaded with GDP (red), GTP (blue), or GppNHp ((a non-hydrolyzable GTP analogue; yellow). (**F**) Hyperbolic dependence of the observed rate constants for mantGDP release from 500 nM *Cr*Arl3 on *Cr*Arl3B·GTP or *Cr*Arl13B·GDP concentration. Fluorescence changes in time at each concentration of *Cr*Arl13B were fitted to single exponentials, and the resulting rate constants ($k_{obs}$) plotted against GEF concentration. $K_{obs}$ values are summarized in **Table 1**.

Next, we employed x-ray crystallography to elucidate the structural basis for the interaction. We thus co-crystallized *Cr*Arl13B·GppNHp and *Cr*Arl3·GDP in the presence of alkaline phosphatase in order to allow formation of nucleotide free *Cr*Arl3. Since *Cr*Arl13B requires bound nucleotide for stability (and most likely for activity) complex formation could not be performed with nucleotide free

**Table 1.** $K_{obs}$ values from data shown in **Figure 2B and E**.

| Concentration dependency (*Figure 2B*) | $K_{obs}$ (s$^{-1}$) ± S.E. |
|---|---|
| *Cr*Arl3 wt intrinsic | $1.2 \times 10^{-4} \pm 1 \times 10^{-5}$ |
| + 0.25 µM *Cr*Arl13BGTP | $1.3 \times 10^{-3} \pm 2 \times 10^{-5}$ |
| + 0.5 µM *Cr*Arl13BGTP | $2.7 \times 10^{-3} \pm 3 \times 10^{-5}$ |
| + 5 µM *Cr*Arl13BGTP | $0.85 \times 10^{-2} \pm 1 \times 10^{-4}$ |
| Nucleotide dependency (*Figure 2E*) | $K_{obs}$ (s$^{-1}$) ± S.E. |
| *Cr*Arl3 wt intrinsic | $1.3 \times 10^{-4} \pm 4 \times 10^{-6}$ |
| + 5 µM *Cr*Arl13BGDP | $9.0 \times 10^{-4} \pm 2 \times 10^{-5}$ |
| + 5 µM *Cr*Arl13BGTP | $0.6 \times 10^{-2} \pm 8 \times 10^{-5}$ |
| + 5 µM *Cr*Arl13BGNP | $0.78 \times 10^{-2} \pm 1 \times 10^{-4}$ |

$K_{obs}$ values ± standard error (S.E.) were determined by fitting the data to single exponential functions.

Arl3 GEF-substrate. The obtained crystals diffracted to 2.5Å and the structure was solved by molecular replacement showing one complex in the asymmetric unit (*Table 3*). Although crystallization was done in the presence of alkaline phosphatase GppNHp was clearly visible in both active sites, suggesting that the structure represents the post-nucleotide-exchange state. The nucleotide dependency of *Cr*Ar13B's GEF activity suggested that switch I and II contribute to the interface. The structure shows indeed that a major part of the interaction is mediated by switch I and II of *Cr*Arl13B which contact *Cr*Arl3 via α4$^{Arl3}$, β6$^{Arl3}$, and α5$^{Arl3}$ located opposite to the nucleotide binding site (*Figure 3A, 3E*). Further interactions are between the long α-helix α6$^{Arl13B}$ which makes a 90° turn at residue G189, and α3/ α4$^{Arl3}$. The last 58 residues, predicted to be α-helical, are not visible in the electron density presumably because they are flexible.

To examine the interface we mutated residues within switch I, II, and α6 of *Cr*Arl13B (*Figure 3B–D*). Switch I mutant *Cr*Arl13B$^{F53A}$ showed a markedly decreased GEF activity whereas the D46A, F51A, N75A, and Y83A mutants had only a minor effect (*Figure 4A*). The charge-reversal mutations K210E/R216E in α6$^{Arl13B}$ as well as D103R and D146R in *Cr*Arl3 show no activity, as expected, whereas a control mutation H154W outside the interface has no effect (*Figure 4B*). Since *Cr*Arl13B's analogous Joubert mutation R77Q and to a lesser extent R194C have been shown to impair the conformational stability of switch II (*Miertzschke et al., 2014*), we next tested the analogous mutants *Cr*Arl13B$^{R77Q}$ and *Cr*Arl13B$^{R194C}$ for their GEF activities. *Cr*Arl13B$^{R77Q}$ displayed a reduced activity in contrast to a very mild effect of *Cr*Arl13B$^{R194C}$ (*Figure 4C*).

GEF proteins normally act by directly interfering with the nucleotide binding site thereby decreasing nucleotide affinity (*Cherfils and Zeghouf, 2013*). In the crystal structure the nucleotide binding site of *Cr*Arl3 is not directly contacted by *Cr*Arl13B. We were not able to trap the interacting residues presumably due to the presence of nucleotide and/or the flexibility of the interacting residues of Arl3B. Considering the length of the C-terminus required for catalysis (see below) it is however suggestive that the mobile C-terminus of Arl13B is involved in the GEF reaction by contacting the relevant surface of Arl3. To examine the importance of this region for catalysis we prepared deletion

**Table 2.** $k_{obs}$ values for the nucleotide dissociation of CrArl3·mGDP and CrArl3·mGppNHp in the presence of CrArl13B·GTP.

| CrArl3·mGDP vs mGppNHp | $K_{obs}$ (s$^{-1}$) ± S.E. |
|---|---|
| *Cr*Arl3·mGDP intrinsic | $1.3 \times 10^{-4} \pm 2 \times 10^{-6}$ |
| *Cr*Arl3·mGDP + 5 µM *Cr*Arl13B·GTP | $0.84 \times 10^{-2} \pm 7 \times 10^{-5}$ |
| *Cr*Arl3·mGppNHp intrinsic | $1.3 \times 10^{-4} \pm 2 \times 10^{-6}$ |
| *Cr*Arl3·mGppNHp + 5 µM *Cr*Arl13B·GTP | $1.0 \times 10^{-2} \pm 1 \times 10^{-4}$ |

$k_{obs}$ rates determined from GEF assays with 0.5 µM *Cr*Arl3 loaded with either mantGDP or mantGppNHp in the presence of 5 µM *Cr*Arl13B$^{18—278}$·GTP and 800 µM unlabeled nucleotide.

**Table 3.** Data collection and refinement statistics (molecular replacement).

| | *Cr*Arl13B-*Cr*Arl3 (5DI3) |
|---|---|
| **Data collection** | |
| Space group | P212121 |
| Cell dimensions | |
| *a, b, c* (Å) | 57.10, 68.80, 120.00 |
| α, β, γ (°) | 90.00, 90.00, 90.00 |
| Resolution (Å) | 29.84 – 2.50 (2.60-2.50) |
| $R_{merge}$ | 0.07 (0.68) |
| $I / \sigma I$ | 17.56 (3.26) |
| Completeness (%) | 99.9 (99.9) |
| Redundancy | 6.4 (6.8) |
| **Refinement** | |
| Resolution (Å) | 2.50 |
| No. reflections | 16944 (1840) |
| $R_{work}/R_{free}$ | 0.199/0.236 |
| No. atoms | 2995 |
| Protein | 2900 |
| Ligand/ion | 2 $Mg^{2+}$, 2 GMPPNP |
| Water | 29 |
| *B*-factors | 66 |
| Protein | 66.40 |
| Ligand/ion | 54.30 |
| Water | 55.00 |
| R.m.s. deviations | |
| Bond lengths (Å) | 0.005 |
| Bond angles (°) | 1.02 |

*Values in parentheses are for highest-resolution shell.

constructs with differing length of the α6-helix (see red asterisks in *Figure 3A*). Whereas the C-terminal deletion constructs △233(18–232) and △243(18–242) had no effect, a longer deletion to residue 220(18–219) showed a reduced stimulation (*Figure 4D*). Finally, the GEF activity of △213(18–212) and △203(18–202) was completely abolished. In support of their importance residues 212–228 are highly conserved among species and we would speculate that these residues contact Arl3 close to the nucleotide binding site.

We next decided to demonstrate the GEF activity of Arl13B in mammalian cells. Therefore we used a stably transfected murine inner medullary collecting duct 3 (IMCD3) cell line and transiently transfected HEK293 cells overexpressing human Arl13B-GFP. To quantify Arl3 activation, Arl3·GTP was affinity-precipitated with the effector GST-PDE6δ (*Linari et al., 1999*) and analysed by immunoblot. The level of endogenous Arl3·GTP was strikingly increased in cells overexpressing Arl13B compared to control cells (*Figure 5A*). Furthermore, the level of GTP-bound Arl3-Flag depended on the Arl13B concentration (*Figure 5B*). Consistent with the Y2H data the Arl2·GTP level was not affected by overexpressed Arl13B indicating selectivity for Arl3 (*Figure 5G*). Interface mutations in Arl13B which disrupted the in vitro exchange activity were also tested in HEK293 cells. Consistently, cells transfected with Arl13B[K216E/R219E], Arl13B[Y55A] or Arl13B[Y85A] did not markedly increase the Arl3·GTP level (*Figure 5C,D*). Intriguingly, the Arl3·GTP level in cells overexpressing the Joubert mutant variants Arl13B[R79Q] and Arl13B[R200C] was lower than those expressing Arl13B[wt]. Consistent with the biochemical data the R79Q mutation impaired Arl3 activation was more pronounced than R200C (*Figure 5E,F*). Finally, we were able to purify human Arl13B (18–278) from insect cells in reasonable

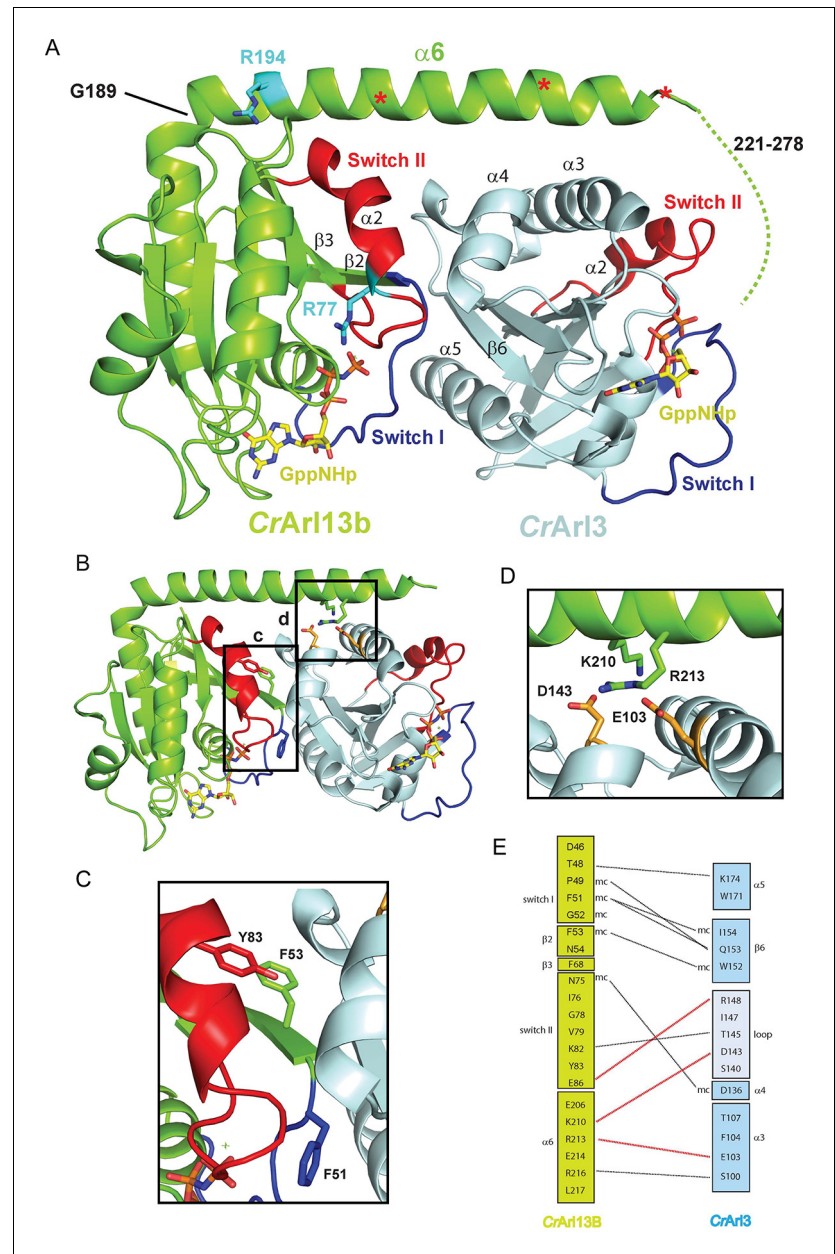

**Figure 3.** The *Cr*Arl13B-–*Cr*Arl3 complex. (**A**) The *Cr*Arl13B-–*Cr*Arl3 complex structure with Arl13B (green), Arl3 (light blue), Switch I (blue), Switch II (red), GppNHp (a non-hydrolyzable (GTP) analogue; yellow). Residues analogous to Joubert syndrome mutations (R77 and R194) are depicted in cyan. Red asterisks delineate the deletion sites (V202, E212, K219) of *Cr*Arl13 used in the guanine nucleotide exchange factor (GEF) assay below (**Figure 4**). Other deletion sites are not resolved in the electron density. Dashed line indicates the 58 C-terminal residues not visible in the structure (**B–D**) Details of the interaction interface. (**C**) Hydrophobic residues located in Switch I and Switch II of *Cr*Arl13B are involved in the interaction with *Cr*Arl3. (**D**) K210 and R213 in α6$^{Arl13B}$ are forming salt bridges with D143$^{Arl3}$ and E103$^{Arl3}$(orange). Coloring as in (**A**). (**E**)Schematic representation of residues located in the interface. Hydrogen bonds between residues are depicted as black dashed line, salt bridges as red dashed line.

amounts to test its GEF activity. Confirming the conservation of structure and function the human Arl13B also exhibits strong GEF activity for Arl3 (**Figure 6A**). The stimulation of the nucleotide release was more efficient compared to *Cr*Arl13B, with a 900fold acceleration at 5 µM. In agreement

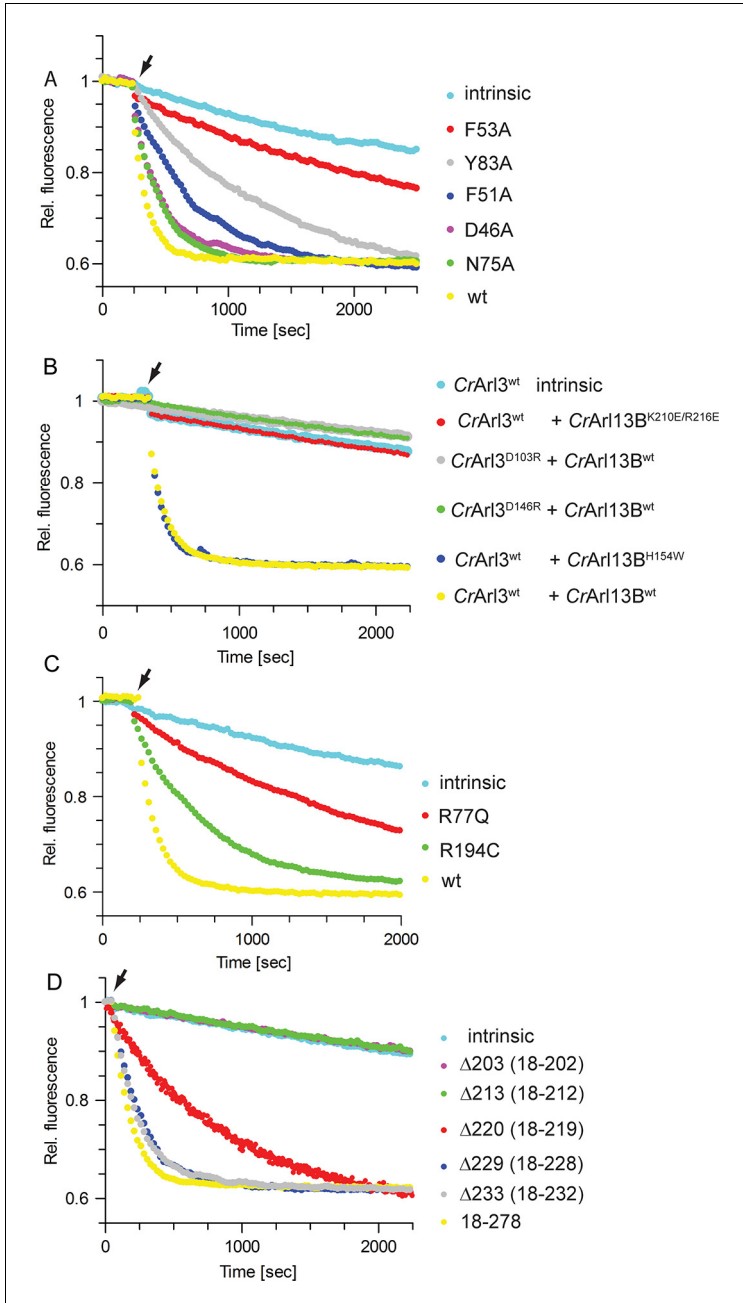

**Figure 4.** Mutations in the CrArl13B-CrArl3 interface and Joubert mutations impair guanine nucleotide exchange factor (GEF) activity. (**A**) GEF activity of CrArl13B$^{18-278}$·GppNHp (a non-hydrolyzable GTP analogue) switch I and II mutants. To CrArl3 mantGppNHp (500 nM) 5 µM of CrArl13B·GppNHp constructs and 800 µM unlabeled GppNHp were added. (**B**) GEF assay with CrArl13B$^{18-278}$·GppNHpand CrArl3·mantGppNHp carrying charge reversal mutations located in the interface. (**C**) GEF activity of the analogous Joubert syndrome mutants (CrArl13B$^{R77Q}$, CrArl13B$^{R194C}$). Same concentrations as in (**A**). (**D**) GEF assay with CrArl13B deletion constructs. Boundaries of deletion fragments: △203: 18–202; △213: 18–213; △220: 18–219; △229: 18–228; △233: 18–232. 18–278 are the constructs used for all other GEF assays. K$_{obs}$ values are summarized in *Table 4*.

with the different biological function (*Zhou et al., 2006*), the nucleotide dissociation of Arl2 was not accelerated by Arl13B (*Figure 6B*).

**Table 4.** $K_{obs}$ values from data shown in **Figure 4 A–D**.

| *Cr*Arl13B Switch interface mutants | $K_{obs}$ (s$^{-1}$) ± S.E. |
|---|---|
| *Cr*Arl3 intrinsic | $1.4 \times 10^{-4} \pm 4 \times 10^{-6}$ |
| + 5 µM *Cr*Arl13B **wt** GTP | $0.91 \times 10^{-2} \pm 2 \times 10^{-4}$ |
| + 5 µM *Cr*Arl13B **F51A** GTP | $2.0 \times 10^{-3} \pm 2 \times 10^{-5}$ |
| + 5 µM *Cr*Arl13B **F53A** GTP | $4.2 \times 10^{-4} \pm 2 \times 10^{-5}$ |
| + 5 µM *Cr*Arl13B **Y83A** GTP | $0.9 \times 10^{-3} \pm 1 \times 10^{-5}$ |
| + 5 µM *Cr*Arl13B **D46A** GTP | $4.1 \times 10^{-3} \pm 8 \times 10^{-5}$ |
| + 5 µM *Cr*Arl13B **N75A** GTP | $4.4 \times 10^{-3} \pm 3 \times 10^{-5}$ |
| *Cr*Arl13B and *Cr*Arl3 Interface mutants | $K_{obs}$ (s$^{-1}$) ± S.E. |
| *Cr*Arl3 **wt** intrinsic | $1.1 \times 10^{-4} \pm 1 \times 10^{-6}$ |
| *Cr*Arl3 **wt** + 5 µM *Cr*Arl13B **K210E/R216E** | $1.5 \times 10^{-4} \pm 1 \times 10^{-5}$ |
| *Cr*Arl3 **D103R** + 5 µM *Cr*Arl13B **wt** | $1.4 \times 10^{-4} \pm 5 \times 10^{-6}$ |
| *Cr*Arl3 **D146R** + 5 µM *Cr*Arl13B **wt** | $1.4 \times 10^{-4} \pm 6 \times 10^{-6}$ |
| *Cr*Arl3 **wt** + 5 µM *Cr*Arl13B **H154W** | $0.88 \times 10^{-2} \pm 2 \times 10^{-4}$ |
| *Cr*Arl3 **wt** + 5 µM *Cr*Arl13B **wt** | $0.85 \times 10^{-2} \pm 2 \times 10^{-4}$ |
| *Cr*Arl13B Deletion constructs | $K_{obs}$ (s$^{-1}$) ± S.E. |
| *Cr*Arl3 **wt** intrinsic | $1.0 \times 10^{-4} \pm 2 \times 10^{-5}$ |
| + 5 µM *Cr*Arl13B △**203** | $1.0 \times 10^{-4} \pm 8 \times 10^{-6}$ |
| + 5 µM *Cr*Arl13B △**213** | $1.0 \times 10^{-4} \pm 1 \times 10^{-5}$ |
| + 5 µM *Cr*Arl13B △**220** | $1.1 \times 10^{-3} \pm 1 \times 10^{-5}$ |
| + 5 µM *Cr*Arl13B △**243** | $4.5 \times 10^{-3} \pm 5 \times 10^{-5}$ |
| + 5 µM *Cr*Arl13B △**233** | $5.0 \times 10^{-3} \pm 6 \times 10^{-5}$ |
| + 5 µM *Cr*Arl13B **18-278** | $6.6 \times 10^{-3} \pm 2 \times 10^{-4}$ |
| *Cr*Arl13B Joubert mutants | $K_{obs}$ (s$^{-1}$) ± S.E. |
| CrArl3 intrinsic | $1.4 \times 10^{-4} \pm 3 \times 10^{-6}$ |
| + 5 µM CrArl13B **R77Q** | $5.5 \times 10^{-4} \pm 1 \times 10^{-5}$ |
| + 5 µM CrArl13B **R194C** | $2.0 \times 10^{-3} \pm 2 \times 10^{-5}$ |
| + 5 µM CrArl13B **wt** | $0.72 \times 10^{-2} \pm 1 \times 10^{-4}$ |

$K_{obs}$ values were determined by fitting the data (**Figure 4 A-D**) to single exponential functions. If not stated otherwise *Cr*Arl13B 18-278 is used for the measurements.

## Discussion

Arl13B has been implicated in a number of ciliary functions (*Cevik et al., 2010*; *Humbert et al., 2012*; *Larkins et al., 2011*; *Li et al., 2010*), and its deletion is causing multiple phenotypes such as the lethal hennin mouse mutant or the scorpion zebrafish mutant (*Caspary et al., 2007*; *Sun et al., 2004*). Here we describe a molecular function for Arl13B acting as GEF for Arl3 whereby the nucleotide state of Arl13B determines its catalytic activity in this activation cascade.

Our results have important implications for the regulation of sorting and transport processes into cilia. It has been shown earlier that Arl3 but not Arl2 can release ciliary cargo from the transport proteins PDE6δ and Unc119 (*Ismail et al., 2012, 2011*; *Wright et al., 2011*). One would predict that Arl3, which is enriched in cilia but also in other microtubule dense structures (*Grayson et al., 2002*; *Zhou et al., 2006*), is only activated inside cilia where Arl13B exclusively resides (*Blacque et al., 2005*; *Caspary et al., 2007*; *Duldulao et al., 2009*) in order to avoid release of ciliary prenylated and myristoylated cargo in the cytoplasm, where other cargo such as Ras, RheB, or Src kinases can be released by Arl2. The observation that expression of constitutive active ARL-3 (Q70L/Q72L) in *Lieshmania donovani* and in *Caenorhabditis elegans* resulted in decreased flagellum length and in

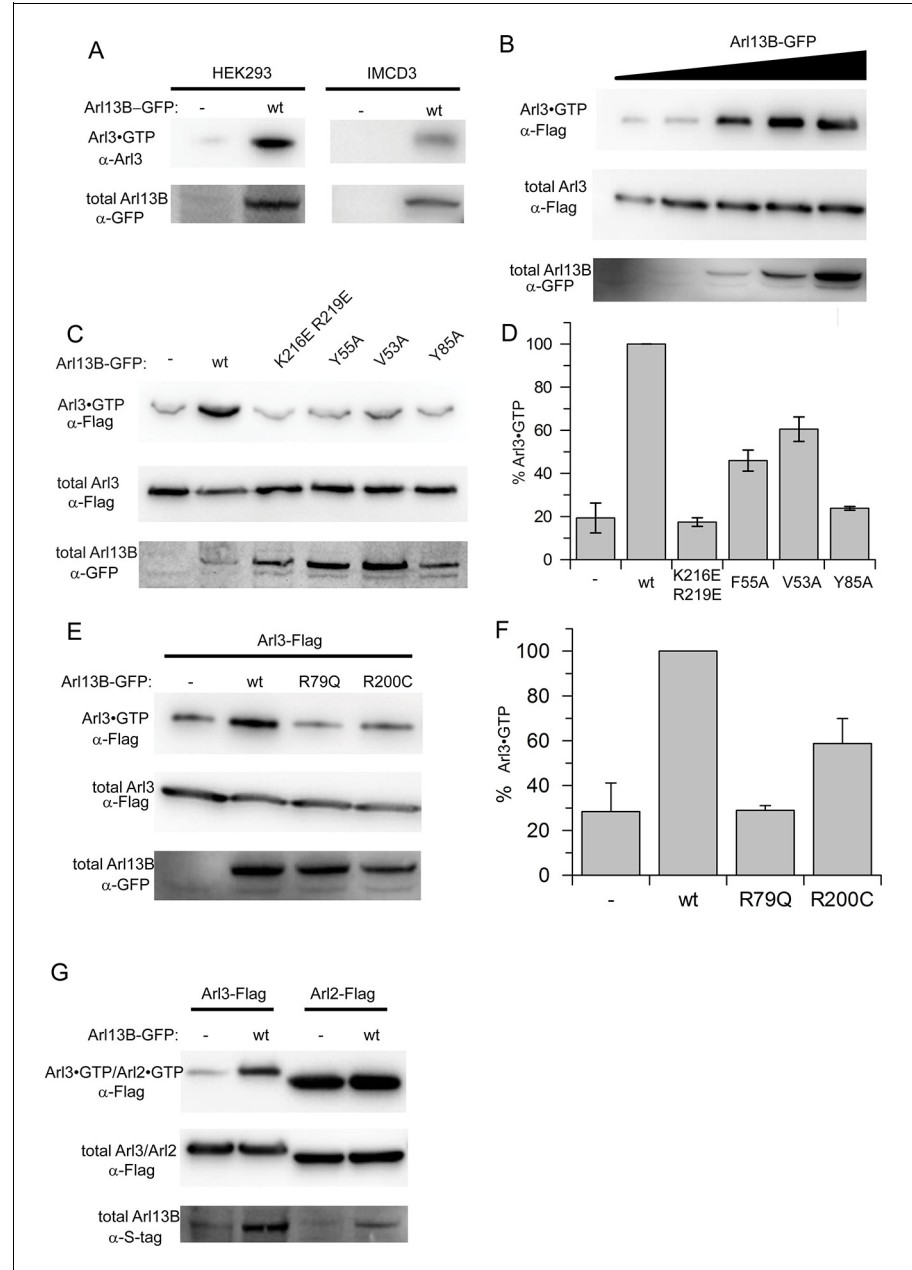

**Figure 5.** Arl13B activates Arl3 in mammalian cells. (**A**) Endogenous Arl3·GTP was affinity-precipitated from Human Embryonic Kidney 293 (HEK293) or murine inner medullary collecting duct 3 (IMCD3) cell lysates using GST-PDE6δ and analyzed as described in Materials and methods. HEK293 cells were transiently transfected with full length Arl13B-GFP(pGLAP5); IMCD3 cells stably expressed the same construct. (**B**) HEK293 cells were transiently transfected with increasing amounts of Arl13B-GFP (0, 1, 3, 6, 12 μg DNA) and constant amounts of Arl3-Flag. Arl3·GTP level determined as in (**A**). (**C**) Arl3-Flag activation in the presence of wildtype and interface mutant Arl13B-GFP was determined as in (**A**) and quantified in (**D**). (**E**) Arl3-Flag activation in the presence Arl13B wt and Joubert syndrome mutants R79Q and R200C. (**F**) Quantification of (**E**). Data is represented as mean ± S.E. (**G**) Arl3-Flag and Arl2-Flag activation in the presence of Arl13B-GFP in HEK293 cells.

impaired ciliogenesis might be explained by Arl3-GTP being located all over the cell and by subsequent mistargeting of proteins destined for the cilium (*Cuvillier et al., 2000*; *Li et al., 2010*).

The different subcellular localization of Arl3-GAP and GEF resembles very closely the Ran driven nucleocytoplasmic transport system. Nucleocytoplasmic transport through the nuclear pore is

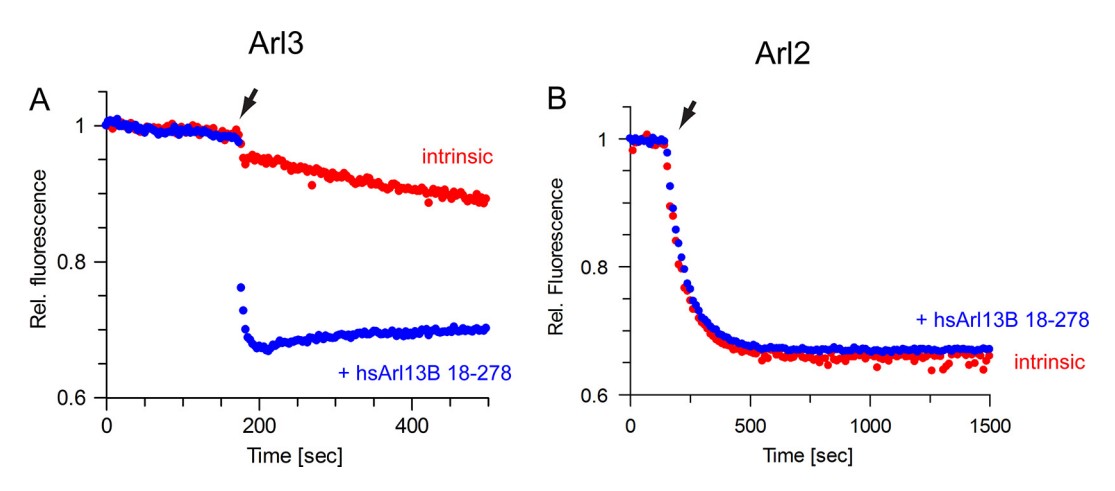

**Figure 6.** The guanine nucleotide exchange factor (GEF) activity of human Arl13B is specific for Arl3. (**A**) GEF activity of human Arl13B[18—278] (purified from insect cells) for murine Arl3. To 500 nM Arl3·mantGppNHp, 5 µM hsArl13B·GTP and 800 µM GTP were added. $k_{obs}$ (intrinsic): $4 \times 10^{-4}$ s$^{-1}$, $k_{obs}$(Arl13B·GTP): 0,36 s$^{-1}$. (**B**) Human Arl13B·GTP does not accelerate nucleotide dissociation of Arl2·mantGppNHp. $k_{obs}$(intrinsic):$1.2 \times 10^{-2}$ s$^{-1}$; $k_{obs}$(Arl13B·GTP): $1.2 \times 10^{-2}$ s$^{-1}$.

regulated by a Ran gradient across the nuclear pore (*Stewart, 2007*). This gradient is regulated by the Ran-GEF RCC1, which is retained inside the nucleus, and by the major form of Ran-GAP, which is located at the exit side of the nuclear pore complex (NPC), by binding to RanBP2 (*Mahajan et al., 1997*). Import cargo bound to importins is released from the carrier by Ran·GTP. The export complex formed by the exportin-cargo complex is in turn stabilized by Ran·GTP and dissociated after exit from the NPC and hydrolysis of GTP. Since the Arl3 specific GAP RP2 is absent from primary cilia and enriched in the preciliary region as observed by us and others (*Blacque et al., 2005*; *Evans et al., 2010*; *Grayson et al., 2002*), we can assume that a similar Arl3·GTP gradient exists across the transition zone and that the Arl3·GTP compartment inside cilia creates a driving force for the transport of prenylated and myristoylated proteins which are allosterically released by Arl3·GTP from their carrier proteins PDE6δ and Unc119a/b (see *Figure 7* for a schematic overview). The Ran-GEF RCC1 is retained in the nucleus through its interaction with nucleosomes (*Nemergut et al., 2001*). In the case of Arl13B, the N-terminal palmitoylation site, but also the other domains seem to be indispensable for its ciliary localization and retention (*Cevik et al., 2010*; *Duldulao et al., 2009*).

Since Arl13B's GEF activity is higher in the GTP-bound conformation one may ask if and how the nucleotide status of Arl13B itself is regulated. We have shown before that the intrinsic GTP hydrolysis activity of Arl13B is very low and that the protein active site does not contain a catalytic glutamine residue (*Miertzschke et al., 2014*). Although we cannot exclude that an Arl13B specific GAP would supply catalytic residues an alternative explanation would be that Arl13B in the absence of GTP hydrolysis is mostly in the GTP-bound form. This does not exclude the existence of an Arl13B-GEF which is presently unknown.

Since both the mutations of the Arl3-GAP RP2 in RP and the Arl3-GEF Arl13B in JSyndrome lead to ciliary defects and ciliopathies, we conclude that the amount of Arl3·GTP needs to be precisely regulated and that both an increase and a decrease of Arl3·GTP is not tolerated for proper function of the cilium.

## Material and methods

### Yeast techniques

Mouse retina cDNA library was generated according to 'Mate&Plate' Library System User Manual (Clonetech), cloned into pGADT7 (short: pAD) and introduced into *Saccharomyces cerevisiae* Y187. Yeast techniques and two-hybrid methods were performed according to the Yeast Protocols Handbook and the Matchmaker GAL4 Two-Hybrid System 3 manual (Clontech) with *S. cerevisiae* AH109.

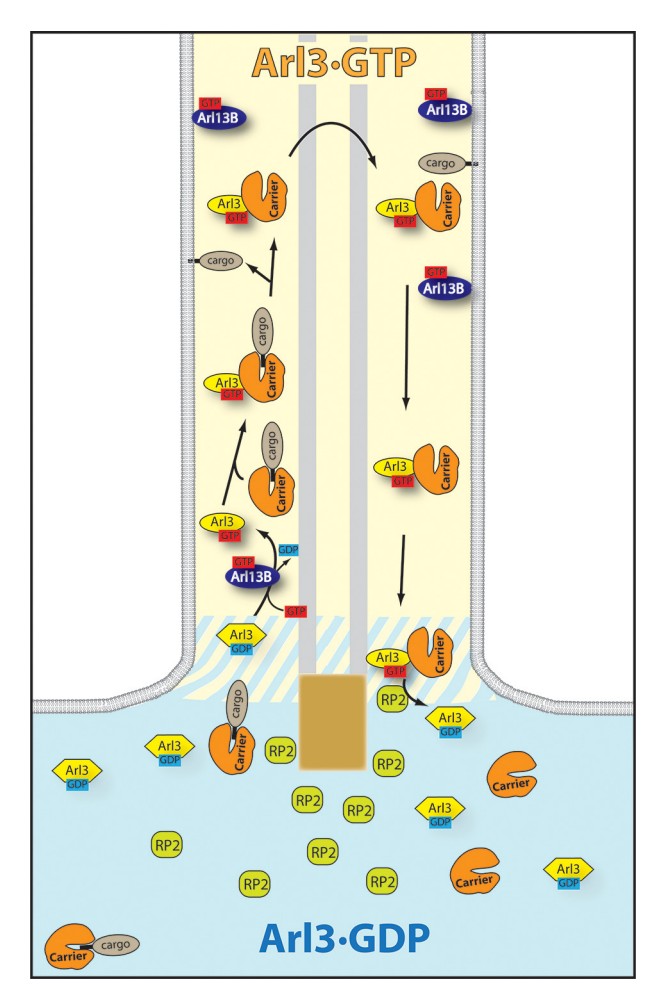

**Figure 7.** The targeting cycle of Arl3 dependent ciliary cargo. In the cilium where Arl13B resides Arl3 gets activated. Through the exclusive localization of Arl13B (Arl3-GEF [guanine nucleotide exchange factor]) inside and retinitis pigmentosa 2 (RP2) (Arl3-GAP) outside the cilium an Arl3·GTP gradient is generated across the transition zone. The carriers PDEδ and Unc119a/b bound to ciliary lipidated cargo reach the cilium where Arl3·GTP binds to the carrier proteins and releases the cargo. RP2 -– enriched in the preciliary region – stimulates the hydrolysis of Arl3·GTP which leads to the dissociation of the carrier proteins from Arl3·GDP.

Murine Arl3△N$^{D129N}$ (residues 17–182), Arl6△N$^{D133N}$ (residues 16–186), Arl2△N$^{D128N}$ (residues 17–184) and Arl13B (residue 20–278)were cloned into a Gateway compatible pBD-Gal4 vector (a kind gift from R. Roepman) and *S. cerevisiae* AH109 used as recipient for transformation.

## Protein expression and purification

*Cr*Arl13B (UniProt: A8INQ0) and *Cr*Arl3 (UniProt: A8ISN6) were amplified by PCR from a cDNA library from *C. reinhardtii* CC-124 WT(wild- type)mt-[137c] [nit1, nit2, agg1] (a gift from T. Happe). Respective mutants were generated by site directed mutagenesis PCR. *Cr*Arl3 and *Cr*Arl13B proteins were expressed as GST-fusions and purified as previously described (*Miertzschke et al., 2014*). *Cr*Arl3 full length was additionally cloned into the pET20 vector to produce C-terminally His-tagged protein. Murine Arl3 full length (UniProt: Q9WUL7) and human Arl2 full length (Uniprot: P36404) in pET20 vectors were already available. Proteins were expressed in BL21DE3 CodonPlus RIL cells at 18°C after induction with 100 µM Isopropyl β-D-1-thiogalactopyranoside (IPTG). Purification of *Cr*Arl3-His, murine Arl3-His and human Arl2-His were conducted as described previously (*Veltel et al., 2008*). Human recombinant His-Arl13B 18–278 was expressed in High-Five

insect cells for 66 hr at 27°C after virus infection. Insect cells were lysed in 30 mM Tris (pH7.5), 150 mM NaCl, 5 mM MgCl$_2$, 3 mM β-mercaptoethanol, 10% glycerole, and 0.1 mM GTP and Complete protease inhibitor cocktail (Roche) using a Microfluidizer M-110S (Microfluidics). Protein was purified by affinity chromatography using a Talon Superflow column (Clonetech) and size exclusion chromatography. All proteins were stored in buffer M containing 25 mM Tris (pH 7.5) 100 mM NaCl, 5 mM MgCl$_2$, 3mM β-mercaptoethanol and 1% glycerole.

## Preparation of proteins with defined nucleotide state

Nucleotide exchange to GDP, GTP, or (N-methylanthraniloyl) mantGDP on Arl proteins was performed in the presence of 50mM ethylenediaminetetraacetic acid (EDTA) and a five fold (two fold for mantGxP) excess of nucleotide. After incubation for 2 hrs 100 mM MgCl$_2$ was added and the protein separated from the excess of nucleotide by a HiTrap desalting column (GE Healthcare). The nucleotide exchange to GppNHp and mantGppNHp was performed using agarose coupled alkaline phosphatase (AP). AP was removed by centrifugation and excess of nucleotide removed by a desalting column. The amount of protein-bound nucleotide was analyzed by C18 reversed-phase high performance liquid chromatography (HPLC) and quantified with a calibrator detector (Beckman Coulter) and an integrator (Shimadzu).

## Pull-down assay with purified protein

Per sample 50 μg GST-$Cr$Arl3 was bound to 50 μl glutathione agarose and washed 2x with 500 μl buffer M. GST-$Cr$Arl3 was incubated in 100 μl buffer M containing 1 mg/ml $Cr$Arl13B$^{18--278}$ (~37 μM) for 30 min and afterwards washed 2x with 500 μl buffer M. Protein was eluted from beads by addition of sodium dodecyl sulfate (SDS) loading buffer and subsequent boiling and analyzed by sodium dodecyl sulfate polyacrylamide gel electrophoresis (SDS-PAGE).

## Crystallization, data collection and analysis

$Cr$Arl13B·GppNHp and $Cr$Arl3·GDP (12 mg/ml) were mixed in the presence of AP in the ratio 1:1.2 (Arl3:Arl13B). With the sitting drop/vapour diffusion method crystals appeared in 0.1M Tris pH 8.5, 25% PEG 6000 (PEGII suite, Qiagen) after 3 days. Crystals were fished out of the 96 well plate and flash frozen in a cryo-solution containing the same constituents as the crystallization condition supplemented with 20% glycerol. Data collection was done at the PXII-XS10SA beamline of the Swiss Light Source (SLS) Villingen. Data were indexed and processed with XDS (*Kabsch, 1993*). Molecular replacement was done with PHASER from the CCP4 package (The CCP4 suite: programs for protein crystallography, 1994). The structure refinement was done using phenix.refine of PHENIX (*Adams et al., 2010*). Images were generated with PYMOL (http://www.pymol.org). Atomic coordinates and structural factors have been deposited in the Protein Data Bank (PDB) under the accession code 5DI3.

## Guanine nucleotide exchange assay

Nucleotide exchange reactions were performed in buffer M at 20°C. As standard conditions, 500 nM G-protein was incubated and the GEF reaction was started with the addition of a mix Arl13B and an excess of nucleotide. Unless otherwise stated 5 μM GEF was used. Since the species of the in excess added unlabeled G-nucleotide (GDP, GTP, or GppNHP) does not influence the velocity of the GEF reaction, the mix always contained an 800-fold excess of the respective nucleotide which was bound to Arl13B in order to avoid undesirable intrinsic nucleotide exchange of Arl13B. For the intrinsic dissociation the same volume buffer containing unlabeled nucleotide was added. The fluorescence change was monitored using a FluoroMax 4 Spectrofluorometer (Jobin Yvon) with an excitation at 366 nm and emission at 450 nm. Data was fitted to single exponential functions using Grafit5 (Erithacus software) to obtain the k$_{off}$ values. All quantitative parameters were measured two or more times. To ensure that all $Cr$Arl13B mutants are 100% loaded with the same nucleotide, they were exchanged to GppNHp with alkaline phosphatase and the stimulation of the nucleotide release measured for $Cr$Arl3·mantGppNHp. K$_M$ and V$_{max}$ were obtained by fitting the data to the Michaelis Menten equation using Grafit5.

## Cells lines

Mouse renal epithelial Flp-In cells from the inner medullary collecting duct (IMCD3 Flp-In; kind gift from MV Nachury) and HEK293 cells were cultured at 37°C and 5% $CO_2$ in Dulbecco's Modified Eagle Medium (DMEM)/F12, 4-(2-hydroxyethyl)-1-piperazineethanesulfonic acid (HEPES) (Life technologies) complemented with 10% fetal bovine serum and 1% L-glutamine.

The parental IMCD3 Flp-In cell line contains a stably integrated FRT cassette and was co-transfected with pOG44 coding a FLP recombinase and the appropriate pgLAP5 vector (Addgene) using Lipofectamine 2000 (Life technologies). For selection of successful stable genomic integration the media was supplemented with 200 µg/ml hygromycin (Merck) and expression of the GFP-fusion protein was checked by Western Blot using an anti-GFP antibody (Santa Cruz Biotechnology).

## Analysis of Arl13B GEF activity in whole cell lysates

For pull-downs of overexpressed Arl3-Flag $2.5 \times 10^6$ HEK293 cells were seeded in 15 cm$^2$ dishes 24 hr prior to transfection. Cells were transfected using Polyethylenimine (PEI) at a ratio 3:1 of PEI (µg) : total DNA (µg). Cells were induced to ciliate by withdrawing serum for 30 hr. ~$2.5 \times 10^7$ cells ($1 \times 15$ cm$^2$ dish) were lysed in 1 ml lysis buffer for 30 min at 4°C. For pull-downs of endogenous Arl3 $1 \times 10^8$ cells ($4 \times 15$ cm$^2$ dish) were used. Lysate was cleared by centrifugation and protein concentration normalized. Per sample 50 µg GST-PDE6$\delta$ was coupled to 50 µl glutathione agarose which was incubated with cleared lysates for 45 min at 4°C. Cleared lysate was removed and beads washed 2x with 500 µl buffer M. Samples were eluted with $1 \times$ SDS-loading buffer. For the detection of affinity-precipitated endogenous Arl3 an anti-Arl3 antibody (Novus Biologicals) was used, and in case of Arl3-Flag an anti-Flag antibody (Thermo Scientific) was used. Expression of Arl13B-GFP was checked using an anti-GFP antibody (Santa Cruz Biotechnology) and antibody against S-peptide, which is located between Arl13B and GFP in pGLAP5. The level of Arl3·GTP was quantified using ImageJ. Experiments were repeated two or more times.

## Acknowledgements

We thank Nadja Schröder for technical help and support to develop the mouse retina cDNA library. We especially thank the Dortmund Protein Facility of the MPI for purifying human Arl13B from insect cells. We are grateful to Raphael Gasper, Susanne Terheyden, Matthias Müller, David Bier, Stefan Baumeister, Ingrid Vetter, and the SLS Beamline scientists Florian Dworkowski, Takashi Tomizaki and Anuschka Pauluhn for data collection at the Swiss Light Source, beamline PXII – X10SA, Paul Scherer Institute, Villigen, Switzerland. We thank Ronald Roepman and Steff Letteboer for providing yeast protocols and helpful suggestions. We thank Stefanie Koesling for the support in the tissue culture and Shehab Ismail for helpful discussions.

## Additional information

### Funding

| Funder | Grant reference number | Author |
| --- | --- | --- |
| European Research Council | ARCID 268782 | Katja Gotthardt<br>Mandy Lokaj<br>Carolin Koerner<br>Alfred Wittinghofer |
| Deutsche Forschungsgemeinschaft | GI 770/1-1 | Nathalie Falk<br>Andreas Gießl |
| Hertha und Helmut Schmauser Stiftung | | Nathalie Falk<br>Andreas Gießl |
| Universitaetsbund Erlangen-Nuernberg E.V. | | Nathalie Falk<br>Andreas Gießl |

The funders had no role in study design, data collection and interpretation, or the decision to submit the work for publication.

## Author contributions

KG, ML, Conception and design, Acquisition of data, Analysis and interpretation of data, Drafting or revising the article; CK, Acquisition of data, Analysis and interpretation of data; NF, AG, Designed and generated the mouse retina cDNA library; AW, Conception and design, Analysis and interpretation of data, Drafting or revising the article

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
