## [Decision Letter]

Thank you for submitting your work entitled "A G-protein activation cascade from Arl13b to Arl3 and implications for ciliary targeting of lipidated proteins" for consideration by *eLife*. Your article has been reviewed by Richard Aldrich (Senior editor), David Clapham (Reviewing editor), and three peer reviewers. One of the three reviewers, DickGörlich (reviewer# 3), has agreed to reveal his identity.

The reviewers have discussed the reviews with one another and the Reviewing editor has drafted this decision to help you prepare a revised submission.

Summary:

The manuscript by Gotthardt et al. identifies Arl13b as the GEF for Arl3 and describes a comprehensive interaction, activity and structural investigation of the two proteins. Arl13b is a G-protein residing in the cilium and is often used as a marker to visualize cilia. However, the molecular function of Arl13b is not known. Arl3 is another G-protein that is required for proper cilium function and is specifically involved in the transport of lipidated cargo proteins to the cilium. The authors of the manuscript carried out yeast-2-hybrid analysis using a mouse retinal cDNA library whereby they identified an interaction between Arl3 and Arl13b. Using recombinantly purified *Chlamydomonas* proteins, for which the solubility was apparently better, it is convincingly shown that Arl13b is indeed a GEF for Arl3. The crystal structure of the complex is also presented and numerous structure-based or disease causing mutations are tested for their impact on GEF activity. The crystal structure represents the post-catalytic state of the complex (Arl3 in a GTP-analog bound state) and thus does not reveal any direct mechanistic insights into the GEF mechanism. The structure does however reveal the molecular basis for Arl3-Arl13b interaction and suggest that residues following a coiled-coil region of Arl13b may be responsible for interaction with the Arl3 G-site, a notion that is confirmed using truncated Arl13b protein in GEF assays. Finally, the manuscript presents the impact of over-expression of WT or mutant Arl13b on the level of Arl3-GTP in different human cell lines and also demonstrates GEF activity for the human proteins pointing to an evolutionarily conserved mechanism.

Essential revision:

1) Figure 2 and corresponding legend are confusing. The legend for C) refers to a red, blue and yellow curve although the curves in panel C) are actually green, blue and yellow. It appears that the figure legends for panel C and E have been swapped. Similarly it appears that the legend for panel D and F have been swapped.

Minor points:

1) Figure 1, Figure 2 and Figure 4: please state whether the numbers following plus/minus standard deviations or standard errors in the legend.

2) Explain the irregularity of the GEF curve in Figure 4 (Y83A, grey curve)? Also, units for the x-axis are missing for Figure 4.

3) The authors state: “In the crystal structure, the nucleotide binding site of *Cr*Arl3 is not directly contacted by *Cr*Arl13B, and cannot be trapped presumably due to the presence of nucleotide and/or its flexibility”. This is quite a bold statement. Maybe change to: “we were not able to trap…”.

4) This sentence needs clarification or rephrasing: “Intriguingly, the Arl3·GTP level of cells overexpressing the Joubert mutant variants Arl13B^R79Q^ and Arl13B^R200C^ was less increased consistent with the biochemical data whereby the effect of R79Q was more drastic (Figure 5)”.

5) Table 3 appears to have a typo for R.m.s.d. for bond length. The number should probably be 0.005̄A instead of 0.0005A. If not then the structure was refined with too tight geometry restraints (even for 0.005A one could argue that it would make sense to loosen the restraints a bit).

6) One reviewer suggested a cartoon illustrating a ciliary targeting cycle.

7) There is an obvious parallel to the RanGTP gradient model for importin and exportin function. Right now, this is just mentioned in passing, late in the Discussion, and not in sufficient detail to grasp the concept. This could be expanded and linked to references. The rationale of the story would become clearer if the Ran-gradient concept were explained first.

8) The main text should start with a coherent introduction. Right now, the introductory paragraph is far too minimalistic for any reader who is not absolutely acquainted with the ciliary targeting field. For example, it should be explained which proteins are targeted, how Arl3 mediates cargo release, how the Arl3 GAP functions in ciliary protein targeting, and why the localisation of this GAP is important. And it should be explained what a ciliopathy phenotype actually is.

These points can be answered in the Discussion:

9) Arl13 itself is a GTP-binding protein, and the authors show that its GTP-bound form has a higher Arl3-GEF activity. How is the nucleotide-bound state of Arl13 controlled? Has Arl3 any effect?

10) It is a bit odd that Arl3 (in the nucleotide exchange intermediate) crystallised with a bound GppNp. Did the authors also try to co-crystallise the D129N mutant? And does this mutant indeed have a higher affinity for Arl13B?

---

## [Author Response]

Essential revision:

*1) Figure 2 and corresponding legend are confusing. The legend for C) refers to a red, blue and yellow curve although the curves in panel C) are actually green, blue and yellow. It appears that the figure legends for panel C and E have been swapped. Similarly it appears that the legend for panel D and F have been swapped.*

This has been corrected.

*Minor points:*

*1) Figure 1, Figure 2 and Figure 4: please state whether the numbers following plus/minus standard deviations or standard errors in the legend.*

Plus/minus standard errors were added to the figures and tables where required.

*2) Explain the irregularity of the GEF curve in Figure 4 (Y83A, grey curve)? Also, units for the x-axis are missing for Figure 4.*

Units for the axis of Figure 4 were fixed. The irregularities in Y83A are an instrumental artefact. All Arl13B mutants behaved well during purification and nucleotide exchange, so that we do not think that the irregularities are derived from instable protein. In the other measurement we did not observe the variations anymore. We exchanged the Y83A curve in Figure 4.

*3) The authors state: “In the crystal structure, the nucleotide binding site of* Cr*Arl3 is not directly contacted by* Cr*Arl13B, and cannot be trapped presumably due to the presence of nucleotide and/or its flexibility”. This is quite a bold statement. Maybe change to: “we were not able to trap…”.*

We changed the sentence accordingly.

4) This sentence needs clarification or rephrasing: “Intriguingly, the Arl3·GTP level of cells overexpressing the Joubert mutant variants Arl13B^R79Q^ and Arl13B^R200C^ was less increased consistent with the biochemical data whereby the effect of R79Q was more drastic (Figure 5)”.

The sentence was rephrased.

5) Table 3 appears to have a typo for R.m.s.d. for bond length. The number should probably be 0.005̄A instead of 0.0005A. If not then the structure was refined with too tight geometry restraints (even for 0.005A one could argue that it would make sense to loosen the restraints a bit).

Table 3 was corrected.

*6) One reviewer suggested a cartoon illustrating a ciliary targeting cycle.*

We included a cartoon showing the Arl3-dependent ciliary targeting cycle.

*7) There is an obvious parallel to the RanGTP gradient model for importin and exportin function. Right now, this is just mentioned in passing, late in the Discussion, and not in sufficient detail to grasp the concept. This could be expanded and linked to references. The rationale of the story would become clearer if the Ran-gradient concept were explained first.*

*8) The main text should start with a coherent introduction. Right now, the introductory paragraph is far too minimalistic for any reader who is not absolutely acquainted with the ciliary targeting field. For example, it should be explained which proteins are targeted, how Arl3 mediates cargo release, how the Arl3 GAP functions in ciliary protein targeting, and why the localisation of this GAP is important. And it should be explained what a ciliopathy phenotype actually is.*

The Discussion and Introduction were extended.

These points can be answered in the Discussion:

*9) Arl13 itself is a GTP-binding protein, and the authors show that its GTP-bound form has a higher Arl3-GEF activity. How is the nucleotide-bound state of Arl13 controlled? Has Arl3 any effect?*

We included a part about the regulation of Arl13B in the Discussion. We analysed if Arl3 has an effect on the nucleotide dissociation of Arl13B (Figure 2), which was not the case. In Figure 2 you can see that the nucleotide dissociation of GTP from Arl13B is slower in the presence of Arl3, most likely due to binding to Arl3. Additionally, we checked if the presence of Arl3 stimulates the GTP hydrolysis of Arl13B. This was not the case either.

*10) It is a bit odd that Arl3 (in the nucleotide exchange intermediate) crystallised with a bound GppNp. Did the authors also try to co-crystallise the D129N mutant? And does this mutant indeed have a higher affinity for Arl13B?*

We tried to purify CrArl3 D129N which unfortunately failed because the protein precipitated during the purification steps. Low affinity mutants of small G proteins very often are unstable after bacterial expression. We also tried to purify the complex of Arl13B-Arl3D129N, but we weren’t able to purify a stable complex either.